# LOCALISED GENERATIVE FLOWS

## ABSTRACT

We argue that flow-based density models based on continuous bijections are limited in their ability to learn target distributions with complicated topologies, and propose *localised generative flows* (LGFs) to address this problem. LGFs are composed of stacked continuous *mixtures* of bijections, which enables each bijection to learn a local region of the target rather than its entirety. Our method is a generalisation of existing flow-based methods, which can be used without modification as the basis for an LGF model. Unlike normalising flows, LGFs do not permit exact computation of log likelihoods, but we propose a simple variational scheme that performs well in practice. We show empirically that LGFs yield improved performance across a variety of density estimation tasks.

## 1 INTRODUCTION

Flow-based generative models, often referred to as *normalising flows*, have become popular methods for density estimation because of their flexibility, expressiveness, and tractable likelihoods. Given the problem of learning an unknown target density $p_X^\star$ on a data space $\mathcal{X}$, normalising flows model $p_X^\star$ as the marginal of $X$ obtained by the generative process

$$Z \sim p_Z, \quad X := g^{-1}(Z), \tag{1}$$

where $p_Z$ is a *prior* density on a space $\mathcal{Z}$, and $g : \mathcal{X} \to \mathcal{Z}$ is a bijection.[1] Assuming sufficient regularity, it follows that $X$ has density $p_X(x) = p_Z(g(x))|\det Dg(x)|$ (see e.g. Billingsley (2008)). The parameters of $g$ can be learned via maximum likelihood given i.i.d. samples from $p_X^\star$.

To be effective, a normalising flow model must specify an expressive family of bijections with tractable Jacobians. Affine coupling layers (Dinh et al., 2014; 2016), autoregressive transformations (Germain et al., 2015; Papamakarios et al., 2017), ODE-based transformations (Grathwohl et al., 2018), and invertible ResNet blocks (Behrmann et al., 2019) are all examples of such bijections that can be composed to produce complicated flows. These models have demonstrated significant promise in their ability to model complex datasets (Papamakarios et al., 2017) and to synthesise novel data points (Kingma & Dhariwal, 2018).

However, in all these cases, $g$ is continuous in $x$. We believe this is a significant limitation of these models since it imposes a *global* constraint on $g^{-1}$, which must learn to match the topology of $\mathcal{Z}$, which is usually quite simple, to the topology of $\mathcal{X}$, which we expect to be very complicated. We argue that this constraint makes maximum likelihood estimation extremely difficult in general, leading to training instabilities and erroneous regions of high likelihood in the learned density landscape.

To address this problem we introduce *localised generative flows* (LGFs), which generalise equation 1 by replacing the single bijection $g$ with stacked continuous mixtures of bijections $\{G(\cdot; u)\}_{u \in \mathcal{U}}$ for an index set $\mathcal{U}$. Intuitively, LGFs allow each $G(\cdot; u)$ to focus on modelling only a *local* component of the target that may have a much simpler topology than the full density. LGFs do not stipulate the form of $G$, and indeed any standard choice of $g$ can be used as the basis of its definition. We pay a price for these benefits in that we can no longer compute the likelihood of our model exactly and must instead resort to a variational approximation, with our training objective replaced by the evidence lower bound (ELBO). However, in practice we find this is not a significant limitation, as the bijective structure of LGFs permits learning a high-quality variational distribution suitable for large-scale training. We show empirically that LGFs outperform their counterpart normalising flows across a variety of density estimation tasks.

---

[1]We assume throughout that $\mathcal{X}, \mathcal{Z} \subseteq \mathbb{R}^d$, and that all densities are with respect to the Lebesgue measure.

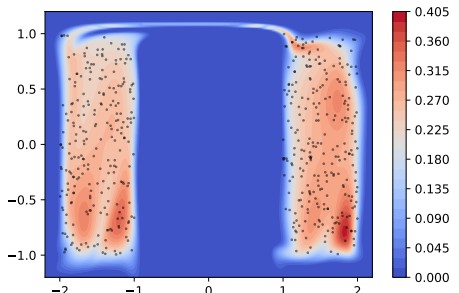 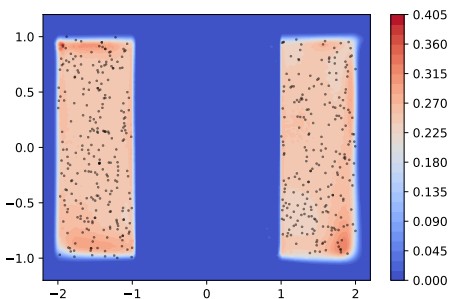

Figure 1: Density models learned after 300 epochs by a standard ten-layer MAF (left) and by a five-layer LGF-MAF (right). Both models have roughly 80,000 parameters and use a standard Gaussian prior. Samples from the target distribution are shown in black. Details of the experimental setup are given in Appendix D.2.

## 2    LIMITATIONS OF NORMALISING FLOWS

Consider a normalising flow model defined by a family of bijections $g_\theta$ parameterised by $\theta$. Suppose we are in the typical case that each $g_\theta$ is continuous in $x$. Intuitively, this seems to pose a problem when $p_X^\star$ and $p_Z$ are supported[2] on domains with different topologies, since continuous bijections (with continuous inverses) necessarily preserve topology. This is made precise in Proposition 1 below; we defer the proof to Appendix A.1. In practice, suggestive pathologies along these lines are readily apparent in simple 2-D experiments as shown in Figure 1. In this case, the density model (a masked autoregressive flow (MAF) (Papamakarios et al., 2017)) is unable to continuously transform the support of the prior (a standard 2-D Gaussian) into the support of $p_X^\star$, which is the union of two disjoint rectangles and hence clearly has a different topology.

**Proposition 1** (Limitations of Normalising Flows). *Suppose each $g_\theta$ is a homeomorphism and $\det Dg_\theta(x) \neq 0$ for all $\theta$ and $x$. If the prior $p_Z$ has a support that is not homeomorphic to $p_X^*$, then no choice of $\theta$ will yield a normalising flow model that matches $p_X^\star$ exactly.*

In other words, once we fix $p_Z$, we immediately restrict the set of target distributions that we may possibly express using our model. Provided each $g_\theta$ is continuous, this occurs no matter how complicated we make our parameterisation – it holds even if we allow *all* continuous bijections. We believe this fact significantly limits the ability of normalising flows to learn any target distribution whose support has a complicated topology.

Furthermore, note that the standard maximum likelihood objective is asymptotically equivalent to

$$\arg \min_\theta D_{\mathrm{KL}}(p_X^\star \| p_X^\theta), \tag{2}$$

where $p_X^\theta(x) = p_Z(g_\theta(x)) |\det Dg_\theta(x)|$ and $D_{\mathrm{KL}}$ denotes the Kullback-Leibler divergence. We conjecture that the problem identified in Proposition 1 is further exacerbated by this objective. Note that equation 2 is infinite unless $\operatorname{supp} p_X^\theta \supseteq \operatorname{supp} p_X^\star$, but strictly positive unless $\operatorname{supp} p_X^\theta = \operatorname{supp} p_X^\star$. Thus equation 2 encourages the support of $p_X^\theta$ to approximate that of $p_X^\star$ from above as closely as possible, but any overshoot – however small – carries an immediate infinite penalty. This seems problematic for gradient-based methods of solving equation 2, especially when, in light of Proposition 1, we expect an optimal $\theta$ to be pathological, since $g_\theta$ is encouraged to approximate some function that is not a continuous bijection. In this context, for near-optimal $\theta$, it seems plausible that $\operatorname{supp} p_X^\theta$ might be very sensitive to perturbations of $\theta$, leading to a highly-varying loss landscape when near the optimum. We conjecture this makes the landscape around the optimal $\theta$ difficult to navigate for gradient-based methods, leading to increasing fluctuations and instabilities

---

[2]As usually defined, the *support* of a density $p$ technically refers to the set on which $p$ is strictly positive. However, our statements are also approximately true when $\operatorname{supp} p$ is interpreted as the region of $p$ which is not smaller than some threshold. This is relevant in practice since, even if both are highly concentrated on some small region, it is common to assume that $p_X^\star$ and $p_Z$ have *full* support, in which case the supports of $p_X^\star$ and $p_Z$ would be trivially homeomorphic.

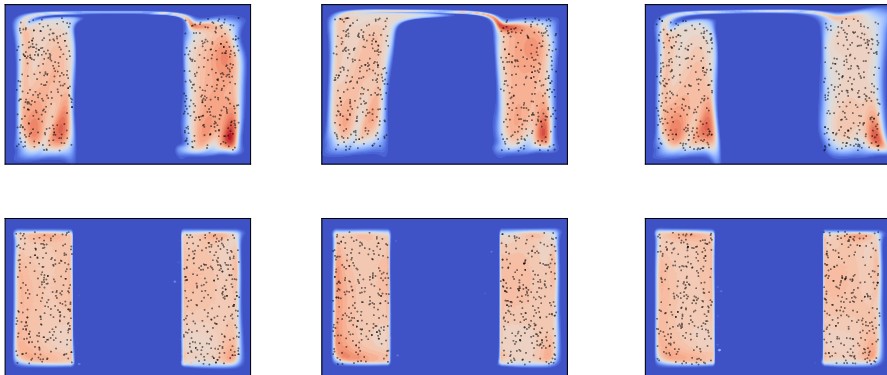

Figure 2: Three subsequent epochs of training for the models shown in Figure 1. The support of the standard MAF (above) fluctuates significantly, while LGF-MAF (below) is stable.

of the KL in equation 2 that degrade the quality of the final $g_\theta$ produced. Figure 2 illustrates this in practice: observe that, even after 300 training epochs, the support of the standard MAF is unstable.

A simple way to fix these problems would be to use a more complicated $p_Z$ that is better matched to the structure of $p_X^\star$. For instance, taking $p_Z$ to be a mixture model has previously been found to improve the performance of normalising flows in some cases (Papamakarios et al., 2017). However, this approach requires prior knowledge of the topology of $\operatorname{supp} p_X^\star$ that might be difficult to obtain: e.g. solving the problem with a Gaussian mixture $p_Z$ requires us to know the number of connected components of $\operatorname{supp} p_X^\star$ beforehand, and even then would require that these components are each homeomorphic to a hypersphere. Ideally, we would like our model to be flexible enough to learn the structure of the target on its own, with minimal explicit design choices required on our part.

An alternative approach would be to try a more expressive family $g_\theta$ in the hope that this better conditions the optimisation problem in equation 2. Several works have considered families of $g_\theta$ that are (in principle) universal approximators of any continuous probability densities (Huang et al., 2018; Jaini et al., 2019). While we have not performed a thorough empirical evaluation of these methods, we suspect that these models can at best mitigate the problems described above, since the assumption of continuity of $g_\theta$ in practice holds for universal approximators also. Moreover, the method we propose below can be used in conjunction with any standard flow, so that we expect even better performance when an expressive $g_\theta$ is combined with our approach.

Finally, we note that the shortcomings of the KL divergence for generative modelling are described at length by Arjovsky et al. (2017). There the authors suggest instead using the Wasserstein distance to measure the discrepancy between $p_X^\star$ and $p_Z$, since under typical assumptions this will yield a continuous loss function suitable for gradient-based training. However, the Wasserstein distance is difficult to estimate in high dimensions, and its performance can be sensitive to the choice of ground metric used (Peyré et al., 2019). Our proposal here is to keep the KL objective in equation 2 and instead modify the model so that we are not required to map $\operatorname{supp} p_Z$ onto $\operatorname{supp} p_X^\star$ using a single continuous bijection. We describe our method in full now.

## 3 LOCALISED GENERATIVE FLOWS

### 3.1 MODEL

The essence of our idea is to replace the single $g$ used in equation 2 with an *indexed family* $\{G(\cdot; u)\}_{u \in \mathcal{U}}$ such that each $G(\cdot; u)$ is a bijection from $\mathcal{X}$ to $\mathcal{Z}$. Intuitively, our aim is for each $G(\cdot; u)$ only to push the prior onto a *local* region of $\operatorname{supp} p_X^\star$, thereby relaxing the constraints posed by standard normalising flows as described above. To do so, we now define $p_X$ as the marginal density of $X$ obtained via the following generative process:

$$Z \sim p_Z, \quad U \sim p_{U|Z}(\cdot|Z), \quad X := G^{-1}(Z; U). \tag{3}$$

Here $p_{U|Z}$ is an additional term that we must specify. In all our experiments we take this to be a mean field Gaussian, so that $p_{U|Z}(u|z) = \operatorname{Gaussian}(u; \mu(z), \sigma(z)^2 I_{d_u})$ for some functions $\mu, \sigma :$

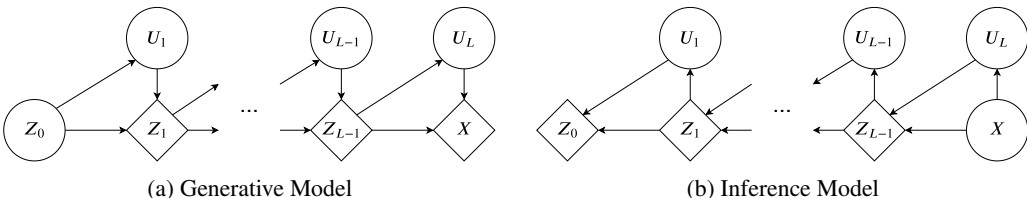

(a) Generative Model        (b) Inference Model

Figure 3: Multi-layer model schematic with $L$ layers. Given their parents, circular nodes are random, and diamond nodes are deterministic.

$\mathcal{Z} \to \mathcal{U} \subseteq \mathbb{R}^{d_u}$, where $d_u$ is the dimension of the index set $\mathcal{U}$, and $I_{d_u}$ is the $d_u \times d_u$ identity matrix. Other possibilities, such as the Concrete distribution (Maddison et al., 2016), might also be useful.

Informally,[3] this yields the joint model $p_{X,U,Z}(x,u,z) = p_Z(z)\,p_{U|Z}(u|z)\,\delta(x - G^{-1}(z;u))$, where $\delta$ is the Dirac delta. We can marginalise out the dependence on $z$ by making the change of variable $z = G(x';u)$, which means $dz = |\det DG(x';u)|\,dx'$.[4] This yields a proper density for $(X,U)$ via

$$p_{X,U}(x,u) = \int p_{X,U,Z}(x,u,z)\,dz = p_Z(G(x;u))\,p_{U|Z}(u|G(x;u))\,|\det DG(x;u)|.$$

We then obtain our density model $p_X$ by integrating over $u$:

$$p_X(x) = \int p_Z(G(x;u))\,p_{U|Z}(u|G(x;u))\,|\det DG(x;u)|\,du. \tag{4}$$

In other words, $p_X(x)$ is a mixture (in general, infinite) of individual normalising flows $G(x;u)$, each weighted by $p_{U|Z}(u|G(x;u))$.

We can also stack this architecture by taking $p_Z$ itself to be a density of the form of equation 4. Doing so with $L$ layers of stacking corresponds to the marginal of $X \equiv Z_L$ obtained via:

$$Z_0 \sim p_{Z_0}, \qquad U_\ell \sim p_{U_\ell|Z_{\ell-1}}(\cdot|Z_{\ell-1}), \qquad Z_\ell = G_\ell^{-1}(Z_{\ell-1};U_\ell), \qquad \ell \in \{1,\dots,L\},$$

where now each $G_\ell(\cdot;u) : \mathcal{X} \to \mathcal{Z}$ is a bijection for all $u \in \mathcal{U}$. The stochastic computation graph corresponding to this model is shown in Figure 3a. In this case, the same argument yields

$$p_{Z_\ell,U_{1:\ell}}(z_\ell,u_{1:\ell}) = p_{Z_{\ell-1},U_{1:\ell-1}}(G_\ell(z_\ell;u_\ell),u_{1:\ell-1})p_{U_\ell|Z_{\ell-1}}(u_\ell|G_\ell(z_\ell;u_\ell))|\det DG_\ell(z_\ell;u_\ell)| \tag{5}$$

where $p_{Z_0,U_{1:0}} \equiv p_{Z_0}$. This approach is in keeping with the standard practice of constructing normalising flows as the composition of simpler bijections, which can indeed be recovered here by taking each $p_{U_\ell|Z_{\ell-1}}(\cdot|Z_{\ell-1})$ to be Dirac. We have found stacking to improve significantly the overall expressiveness of our models, and make extensive use of it in our experiments below.

## 3.2 BENEFITS

Heuristically, we believe our model allows each $G(\cdot;u)$ to learn a *local* region of $p_X^\star$, thereby greatly relaxing the global constraints on existing flow-based models described above. To ensure a finite KL, we no longer require the density $p_Z(G(x;u))\,|\det DG(x;u)|$ to have support covering the entirety of $\operatorname{supp} p_X^\star$ for any given $u$; all that matters is that every region of $\operatorname{supp} p_X^\star$ is covered for *some* $u$. Our model can thus achieve good performance with each bijection $G(\cdot;u)$ faithfully mapping onto a potentially very small component of $p_X^\star$. This seems intuitively more achievable than the previous case, wherein a single bijection is required to capture the entire target.

This argument is potentially clearest if $u$ is discrete. For example, even if $u \in \{u_{-1}, u_1\}$ can take only two possible values, it immediately becomes simpler to represent the target shown in Figure 1 using a Gaussian prior: we simply require $G^{-1}(\cdot;u_{-1})$ to map onto one component, and $G^{-1}(\cdot;u_1)$ to map onto the other. In practice, we could easily implement such a $G$ using two separate normalising flows that are trained jointly. The discrete case is also appealing since in principle it allows exact evaluation of the integral in equation 4, which becomes a summation. Unfortunately this approach also has significant drawbacks that we discuss at greater length in Appendix B.

We therefore instead focus here on a continuous $u$. In this case, for example, we can recover $p_X^\star$ from Figure 1 by partitioning $\mathcal{U}$ into disjoint regions $\mathcal{U}_{-1}$ and $\mathcal{U}_1$, and having $G^{-1}(\cdot;u)$ map onto

---

[3]This argument can be made precise using disintegrations (Chang & Pollard, 1997), but since the proof is mainly a matter of measure-theoretic formalities we omit it.

[4]Note that $DG(x;u)$ refers to the Jacobian with respect to $x$ only.

the left component of $p_X^\star$ for $u \in \mathcal{U}_{-1}$, and the right component for $u \in \mathcal{U}_1$. Observe that in this scenario we do not require any given $G^{-1}(\cdot; u)$ to map onto *both* components of the target, which is in keeping with our goal of localising the model of $p_X^\star$ that is learned by our method.

In practice $G$ will invariably be continuous in both its arguments, in which case it will not be possible to partition $\mathcal{U}$ disjointly in this way. Instead we will necessarily obtain some additional intermediate region $\mathcal{U}_0$ on which $G^{-1}(\cdot; u)$ maps part of $\mathrm{supp}\, p_Z$ outside of $\mathrm{supp}\, p_X^\star$, so that $p_X(x)$ will be strictly positive there. However, Proposition 2 below shows (proof in Appendix A.2) that this is not a problem for LGFs – we are able to use $p_{U|Z}$ to downweight such a region, which avoids the support mismatch issue of standard normalising flows.

**Proposition 2** (Benefit of Localised Generative Flows). *Suppose that* $\mathrm{supp}\, p_X^*$ *is open and* $G(\cdot; u)$ *is continuous for each* $u$. *Suppose further that, for each* $x \in \mathrm{supp}\, p_X^*$, *the set*
$$B_x := \{u \in \mathcal{U} \mid p_Z(G(x; u)) | \det DG(x; u)| > 0\}$$
*has positive Lebesgue measure. Then there exists* $p_{U|Z}$ *such that* $\mathrm{supp}\, p_X = \mathrm{supp}\, p_X^*$.

### 3.3 INFERENCE

Even in the single layer case ($L = 1$), if $u$ is continuous, then the integral in equation 4 is intractable. In order to train our model, we therefore resort to a variational approximation: we introduce an approximate posterior $q_{U_{1:L}|X} \approx p_{U_{1:L}|X}$, and consider the evidence lower bound (ELBO) of $\log p_X(x)$:
$$\mathcal{L}(x) := \mathbb{E}_{q_{U_{1:L}|X}(u_{1:L}|x)} \left[ \log p_{X, U_{1:L}}(x, u_{1:L}) - \log q_{U_{1:L}|X}(u_{1:L}|x) \right].$$
It is straightforward to show that $\mathcal{L}(x) \leq \log p_X(x)$, and that this bound is tight when $q_{U_{1:L}|X}$ is the exact posterior $p_{U_{1:L}|X}$. This allows learning an approximation to $p_X^\star$ by maximising $n^{-1} \sum_{i=1}^n \mathcal{L}(x_i)$ jointly in $p_{X, U_{1:L}}$ and $q_{U_{1:L}|X}$, for a dataset of $n$ i.i.d. samples $x_i \sim p_X^\star$.

It can be shown (see Appendix A.3) that the exact posterior factors as $p_{U_{1:L}|X}(u_{1:L}|x) = \prod_{\ell=1}^L p_{U_\ell|Z_\ell}(u_\ell|z_\ell)$, where $z_L := x$, and $z_{\ell-1} := G_\ell(z_\ell; u_\ell)$ for $\ell \leq L$. We thus endow $q_{U_{1:L}|X}$ with the same form:
$$q_{U_{1:L}|X}(u_{1:L}|x) := \prod_{\ell=1}^L q_{U_\ell|Z_\ell}(u_\ell|z_\ell).$$

The stochastic computation graph for this inference model is shown in Figure 3b. In conjunction with equation 5, this allows writing the ELBO recursively as
$$\mathcal{L}_\ell(z_\ell) = \mathbb{E}_{q_{U_\ell|Z_\ell}(u_\ell|z_\ell)} \left[ \mathcal{L}_{\ell-1}(G_\ell(z_\ell; u_\ell)) + \log p_{U|Z_{\ell-1}}(u_\ell|z_{\ell-1}) + \log |\det DG_\ell(z_\ell; u_\ell)| - \log q_{U_\ell|Z_\ell}(u_\ell|z_\ell) \right]$$
for $\ell \geq 1$, with the base case $\mathcal{L}_0(z_0) = p_{Z_0}(z_0)$. Here we recover $\mathcal{L}(x) \equiv \mathcal{L}_L(z_L)$.

Now let $\theta$ denote all the parameters of both $p_{X, U_{1:L}}$ and $q_{U_{1:L}|X}$. Suppose each $q_{U_\ell|Z_\ell}$ can be suitably reparametrised (Kingma & Welling, 2013; Rezende et al., 2014) so that $h_\ell(\epsilon_\ell, z_\ell) \sim q_{U_\ell|Z_\ell}(\cdot|z_\ell)$ when $\epsilon_\ell \sim \eta_\ell$ for some function $h_\ell$ and density $\eta_\ell$, where $\eta_\ell$ does not depend on $\theta$. In all our experiments we give $q_{U_\ell|Z_\ell}$ the same mean field form as in equation 3, in which case this holds immediately as described e.g. by Kingma & Welling (2013). We can then obtain unbiased estimates of $\nabla_\theta \mathcal{L}(x)$ straightforwardly using Algorithm 1, which in turn allows minimising our objective via stochastic gradient descent. Note that although this algorithm is specified in terms of a single value of $z_\ell$, it is trivial to obtain an unbiased estimate of $\nabla_\theta m^{-1} \sum_{j=1}^m \mathcal{L}(x_j)$ for a minibatch of points $\{x_j\}_{j=1}^m$ by averaging over the batch index at each layer of recursion.

---

**Algorithm 1** Recursive calculation of an unbiased estimator of $\nabla_\theta \mathcal{L}_\ell(z_\ell)$

---

   **function** GRADELBO($z_\ell, \ell$)
      **if** $\ell = 0$ **then**
         **return** $\nabla_\theta \log p_{Z_0}(z_\ell)$
      **else**
         $\epsilon_\ell \sim \eta_\ell$
         $u_\ell \leftarrow h_\ell(\epsilon_\ell, z_\ell)$
         $z_{\ell-1} \leftarrow G_\ell(z_\ell; u_\ell)$
         $\Delta_\ell \leftarrow \log p_{U_\ell|Z_{\ell-1}}(u_\ell|z_{\ell-1}) + \log |\det DG_\ell(z_\ell; u_\ell)| - \log q_{U_\ell|Z_\ell}(u_\ell|z_\ell)$
         **return** GRADELBO($z_{\ell-1}, \ell - 1$) $+ \nabla_\theta \Delta_\ell$

---

### 3.3.1 PERFORMANCE

A major reason for the popularity of normalising flows is the tractability of their exact log likelihoods. In contrast, the variational scheme described here can produce at best an approximation to this value, which we might expect reduces performance of the final density estimator learned. Moreover, particularly when the number of layers $L$ is large, it might seem that the variance of gradient estimators obtained from Algorithm 1 would be impractically high.

However, in practice we have not found either of these problems to be a significant limitation, as our experimental results in Section 5 show. Empirically we find improved performance over standard flows even when using the ELBO as our training objective. We also find that importance sampling is sufficient for obtaining good, low-variance (if slightly negatively biased) estimates of $\log p_X(x)$ (Rezende et al., 2014, Appendix E) at test time, although we do note that the stochasticity here can lead to small artefacts like the occasional white spots visible above in our 2-D experiments.

We similarly do not find that the variance of Algorithm 1 grows intractably when we use a large number of layers $L$, and in practice we are able to train models having the same depth as popular normalising flows. We conjecture that this occurs because, as the number of stacked bijections in our generative process grows, the complexity required of each *individual* bijection to map $p_Z$ to $p_X^\star$ naturally decreases. We therefore have reason to think that, as $L$ becomes large, learning each $q_{U_\ell|Z_\ell}$ will become easier, so that the variance at each layer will decrease and the *overall* variance will remain fairly stable.

### 3.4 CHOICE OF INDEXED BIJECTION FAMILY

We now consider the choice of $G$, for which there are many possibilities. In our experiments, which all take $\mathcal{Z} = \mathbb{R}^d$, we focus on the simple case of

$$G(x; u) = \exp(s(u)) \odot g(x) + t(u), \tag{6}$$

where $s, t : \mathcal{U} \to \mathcal{Z}$ are unrestricted mappings, $g : \mathcal{X} \to \mathcal{Z}$ is a bijection, and $\odot$ denotes elementwise multiplication. In this case $\log|\det DG(x; u)| = \log|\det Dg(x)| + \sum_{i=1}^d [s(u)]_i$, where $[s(u)]_i$ is the $i^{\text{th}}$ component of $s(u)$. This has the advantage of working out-of-the-box with all pre-existing normalising flow methods for which a tractable Jacobian of $g$ is available. We provide alternative suggestions to this construction in Appendix C.

Equation 6 also has an appealing similarity with the common practice of applying affine transformations between flow steps for normalisation purposes, which has been found empirically to improve stability, convergence time, and overall performance (Dinh et al., 2016; Papamakarios et al., 2017; Kingma & Dhariwal, 2018). In prior work, $s$ and $t$ have been simple parameters that are learned either directly as part of the model, or updated according to running batch statistics. Our approach may be understood as a generalisation of these techniques.

## 4 RELATED WORK

### 4.1 DISCRETE MIXTURE METHODS

Several density models exist that involve discrete mixtures of normalising flows. A closely-related approach to ours is RAD (Dinh et al., 2019), which is a special case of our model when $L = 1$, $u$ is discrete, and $\mathcal{X}$ is partitioned disjointly. In the context of Monte Carlo estimation, Duan (2019) proposes a similar model to RAD that does not use partitioning. Ziegler & Rush (2019) introduce a normalising flow model for sequences with an additional latent variable indicating sequence length. More generally, our method may be considered an addition to the class of *deep mixture models* (Tang et al., 2012; Van den Oord & Schrauwen, 2014), with our use of continuous mixing variables designed to reduce the computational difficulties that arise when stacking discrete mixtures hierarchically – we refer the reader to Appendix B for more details on this.

### 4.2 METHODS COMBINING VARIATIONAL INFERENCE AND NORMALISING FLOWS

There is also a large class of methods which use normalising flows to improve the inference procedure in variational inference (van den Berg et al., 2018; Kingma et al., 2016; Rezende & Mohamed, 2015), although flows are not typically present in the generative process here. This approach can be described as *using normalising flows to improve variational inference*, which contrasts our goal of *using variational inference to improve normalising flows*.

However, there are indeed some methods augmenting normalising flows with variational inference, but in all cases below the variational structure is not stacked to obtain extra expressiveness. Ho et al. (2019) use a variational scheme to improve upon the standard dequantisation method for deep generative modelling of images (Theis et al., 2015); this approach is orthogonal to ours and could indeed be used alongside LGFs. Gritsenko et al. (2019) also generalise normalising flows using variational methods, but they incorporate the extra latent noise into the model in a much more restrictive way. Das et al. (2019) only learn a low-dimensional prior over the noise space variationally.

### 4.3 STACKED VARIATIONAL METHODS

Finally, we can contrast our approach with purely variational methods which are not flow-based, but still involve some type of stacking architecture. The main difference between these methods and LGFs is that the bijections we use provide us with a generative model with far more structure, which allows us to build appropriate inference models more easily. Contrast this with, for example, Rezende et al. (2014), in which the layers are independently inferred, or Sønderby et al. (2016), which requires a complicated parameter-sharing scheme to reliably perform inference. A single-layer instance of our model also shares some similarity with the fully-unsupervised case in Maaløe et al. (2016), but the generative process there conditions the auxiliary variable on the data.

## 5 EXPERIMENTS

We evaluated the performance of LGFs on several problems of varying difficulty, including synthetic 2-D data, several UCI datasets, and two image datasets. We describe the results of the UCI and image datasets here; Appendix D.2 contains the details of our 2-D experiments.

In each case, we compared a baseline flow to its extension as an LGF model with roughly the same number of parameters. We obtained each LGF by inserting a mixing variable $U_\ell$ between every component layer of the baseline flow. For example, we inserted a $U_\ell$ after each autoregressive layer in MAF (Papamakarios et al., 2017), and after each affine coupling layer in RealNVP (Dinh et al., 2016). In all cases we obtained $p_{U_\ell|Z_{\ell-1}}$, $q_{U_\ell|Z_\ell}$, $s_\ell$, and $t_\ell$ as described in Appendix D.1. We inserted batch normalisation (Ioffe & Szegedy, 2015) between flow steps for the baselines as suggested by Dinh et al. (2016), but omit it for the LGFs, since our choice of bijection family is a generalisation of batch normalisation as described in Subsection 3.4.

We trained our models to maximise either the log-likelihood (for the baselines) or the ELBO (for the LGFs) using the ADAM optimiser (Kingma & Ba, 2014) with default hyperparameters and no weight decay. For the UCI and image experiments we stopped training after 30 epochs of no validation improvement for the UCI experiments, and after 50 epochs for the image experiments. Both validation and test performance were evaluated using the exact log-likelihood for the baseline, and the standard importance sampling estimator of the average log-likelihood (Rezende et al., 2014, Appendix E) for LGFs. For validation we used 5 importance samples for the UCI datasets and 10 for the image datasets, while for testing we used 1000 importance samples in all cases. Our code is available at `https://github.com/anonsubmission974/lgf`.

### 5.1 UCI DATASETS

We tested the performance of LGFs on the POWER, GAS, HEPMASS, and MINIBOONE datasets from the UCI repository (Bache & Lichman, 2013). We preprocessed these datasets identically to Papamakarios et al. (2017), and use the same train/validation/test splits. For all models we used a batch size of 1000 and a learning rate of $10^{-3}$. These constitute a factor of 10 increase over the values used by Papamakarios et al. (2017) and were chosen to decrease training time. Our baseline results differ slightly from Papamakarios et al. (2017), which may be as a result of this change.

We focused on MAF as our baseline normalising flow because of its improved performance over alternatives such as RealNVP for general-purpose density estimation (Papamakarios et al., 2017). A given MAF model is defined by how many autoregressive layers it uses as well as the sizes of the autoregressive networks at each layer. For an LGF-MAF, we must additionally define the neural networks used in the generative and inference process, for which we use multi-layer perceptrons (MLPs). We considered a variety of choices of hyperparameters for MAF and LGF-MAF. Each instance of LGF-MAF had a corresponding MAF configuration, but in order to compensate for the parameters introduced by our additional neural networks, we also considered deeper and wider MAF models. For each LGF-MAF, the dimensionality of $u_\ell$ was roughly a quarter of that of the data. Full hyperparameter details are given in Appendix D.3. For three random seeds, we trained models using

Table 1: Average plus/minus standard error of the best test-set log-likelihood.

|  | **POWER** | **GAS** | **HEPMASS** | **MINIBOONE** |
|---|---|---|---|---|
| MAF | $0.19 \pm 0.02$ | $9.23 \pm 0.07$ | $-18.33 \pm 0.10$ | $-10.98 \pm 0.03$ |
| LGF-MAF | $\mathbf{0.48 \pm 0.01}$ | $\mathbf{12.02 \pm 0.10}$ | $\mathbf{-16.63 \pm 0.09}$ | $\mathbf{-9.93 \pm 0.04}$ |

every hyperparameter configuration, and then chose the best-performing model across *all* parameter configurations using validation performance. Table 1 shows the resulting test-set log-likelihoods averaged across the different seeds. It is clear that LGF-MAFs yield improved results in this case.

## 5.2 IMAGE DATASETS

We also considered LGFs applied to the Fashion-MNIST (Xiao et al., 2017) and CIFAR-10 (Krizhevsky et al., 2009) datasets. In both cases we applied the dequantisation scheme of Theis et al. (2015) beforehand, and trained all models with a learning rate of $10^{-4}$ and a batch size of 100.

We took our baseline to be a RealNVP with the same architecture used by Dinh et al. (2016) for their CIFAR-10 experiments. In particular, we used 10 affine coupling layers with the corresponding alternating channelwise and checkerboard masks. Each coupling layer used a ResNet (He et al., 2016a;b) consisting of 8 residual blocks of 64 channels (denoted $8 \times 64$ for brevity). We also replicated their multi-scale architecture, squeezing the channel dimension after the first 3 coupling layers, and splitting off half the dimensions after the first 6. This model had 5.94M parameters for Fashion-MNIST and 6.01M parameters for CIFAR-10. For completeness, we also consider a RealNVP model with coupler networks of size $4 \times 64$ to match those used below in LGF-RealNVP. This model had 2.99M and 3.05M parameters for Fashion-MNIST and CIFAR-10 respectively.

For the LGF-RealNVP, we sought to maintain roughly the same depth over which we propagate gradients as in the baseline. To this end, our coupling networks were ResNets of size $4 \times 64$, and each $p_{U_\ell | Z_{\ell-1}}$ and $q_{U_\ell | Z_\ell}$ used ResNets of size $2 \times 64$. Our $(s_\ell, t_\ell)$ network was a ResNet of size $2 \times 8$. We give $u_\ell$ the same shape as a single channel of $z_\ell$, and upsampled

Table 2: Average plus/minus standard error of test-set bits per dimension. RealNVP ($k$) refers to a RealNVP model with $k$ residual blocks in the coupling networks.

|  | **Fashion-MNIST** | **CIFAR-10** |
|---|---|---|
| RealNVP (4) | $2.944 \pm 0.003$ | $3.565 \pm 0.001$ |
| RealNVP (8) | $2.946 \pm 0.002$ | $3.554 \pm 0.001$ |
| LGF-RealNVP (4) | $\mathbf{2.823 \pm 0.003}$ | $\mathbf{3.477 \pm 0.019}$ |

to the dimension of $z_\ell$ by adding channels at the output of the $(s_\ell, t_\ell)$ ResNet. Our model had 5.99M parameters for Fashion-MNIST and 6.07M parameters for CIFAR-10.

Table 2 shows that LGFs consistently outperformed the baseline models. We moreover found that LGFs tend to train *faster*: the average epoch with best validation performance on CIFAR-10 was 458 for LGF-RealNVP, and 723 for RealNVP (8).[5] Samples from all models are shown in Appendix D.4.

We also found that using the ELBO instead of the log-likelihood does not penalise our method. The gap between the estimated test-set log-likelihood and the average test-set ELBO was not very large for the LGF models, with a relative error of $8.98 \times 10^{-3}$ for Fashion-MNIST and $6.88 \times 10^{-3}$ for CIFAR-10. Moreover, the importance-sampling-based log-likelihood estimator itself had very low variance when using 1000 importance samples. For each trained LGF model, we estimated the relative standard deviation using three separate samples of this estimator. We obtained an average over all trained models of $8.34 \times 10^{-4}$ for Fashion-MNIST, and $2.07 \times 10^{-5}$ for CIFAR-10.

## 6 CONCLUSION

In this paper, we have proposed localised generative flows for density estimation, which generalise existing normalising flow methods and address the limitations of these models in expressing complicated target distributions. Our method obtains successful empirical results on a variety of tasks. Many extensions appear possible, and we believe localised generative flows show promise as a means for improving the performance of density estimators in practice.

---

[5]No RealNVP (4) run had converged after 1000 epochs, at which point we stopped training. However, by this point the rate of improvement had slowed significantly.

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

# A  ADDITIONAL PROOFS

## A.1  PROOF OF PROPOSITION 1 (SEE PAGE 2)

**Proposition 1** (Limitations of Normalising Flows). *Suppose each $g_\theta$ is a homeomorphism and* $\det Dg_\theta(x) \neq 0$ *for all $\theta$ and $x$. If the prior $p_Z$ has a support that is not homeomorphic to $p_X^\star$, then no choice of $\theta$ will yield a normalising flow model that matches $p_X^\star$ exactly.*

*Proof.* For any choice of $\theta$, let $p_X^\theta(x)$ be the normalising flow model produced by $p_Z$ and $g_\theta$, i.e.
$$p_X^\theta(x) = p_Z(g_\theta(x))|\det Dg_\theta(x)|.$$
We can bound the total variation distance $d_{\mathrm{TV}}$ between $p_X^\theta$ and $p_X^\star$ by

$$d_{\mathrm{TV}}(p_X^\star, p_X^\theta) \geq \left| \int_{\mathrm{supp}\, p_X^\star} p_X^\star(x)dx - \int_{\mathrm{supp}\, p_X^\star} p_X^\theta(x)dx \right|$$

$$= \left| 1 - \int_{\mathrm{supp}\, p_X^\star \cap \mathrm{supp}\, p_X^\theta} p_X^\theta(x)dx \right|.$$

Observe that this is strictly positive unless $\mathrm{supp}\, p_X^\star = \mathrm{supp}\, p_X^\theta$ (up to a set of $p_X^\theta$-probability 0). But since $g_\theta$ is a homeomorphism, this is not possible. Hence $d_{\mathrm{TV}}(p_X^\star, p_X^\theta) > 0$, so that $p_X^\star \neq p_X^\theta$. $\quad\square$

## A.2  PROOF OF PROPOSITION 2 (SEE PAGE 5)

**Proposition 2** (Benefit of Localised Generative Flows). *Suppose that $\mathrm{supp}\, p_X^*$ is open and $G(\cdot; u)$ is continuous for each $u$. Suppose further that, for each $x \in \mathrm{supp}\, p_X^*$, the set*
$$B_x := \{u \in \mathcal{U} \mid p_Z(G(x; u))|\det DG(x; u)| > 0\}$$
*has positive Lebesgue measure. Then there exists $p_{U|Z}$ such that $\mathrm{supp}\, p_X = \mathrm{supp}\, p_X^*$.*

*Proof.* Take $p_U$ to be any positive density. Define
$$A_u := G(\mathrm{supp}\, p_X^*; u).$$
Observe that each $A_u$ has positive Lebesgue measure, since it is the continuous injective image of an open set. Thus, for each $u$, we can choose $p_{Z|U}(\cdot|u)$ to be any positive density on $A_u$. This allows us to define
$$p_{U|Z}(u|z) := \frac{p_{Z|U}(z|u)p_U(u)}{\int p_{Z|U}(z|u')p_U(u')du'}.$$

We claim this construction gives $\mathrm{supp}\, p_X = \mathrm{supp}\, p_X^*$. First suppose $x \in \mathrm{supp}\, p_X^*$. For each $u$, observe that $p_{Z|U}(G(x; u)|u) > 0$ by the definition of $A_u$, and hence $p_{U|Z}(u|G(x; u)) > 0$ since $p_U$ is positive. Consequently,

$$p_X(x) \geq \int_{B_x} p_Z(G(x; u))p_{U|Z}(u|G(x; u))|\det DG(x; u)|\, du > 0$$

since the integrand is strictly positive on $B_x$, which has positive measure.

Now suppose $x \notin \mathrm{supp}\, p_X^*$. For each $u$, since $p_{Z|U}(\cdot|u)$ is supported on $A_u$, we must have $p_{Z|U}(G(x; u)|u) = 0$, and hence $p_{U|Z}(u|G(x; u)) = 0$. From this it follows directly that $p_X(x) = 0$, i.e. $x \notin \mathrm{supp}\, p_X$. $\quad\square$

## A.3  CORRECTNESS OF POSTERIOR FACTORISATION

We want to prove that the exact posterior factorises according to

$$p_{U_{1:L}|X}(u_{1:L}|x) = \prod_{\ell=1}^{L} p_{U_\ell|Z_\ell}(u_\ell|z_\ell), \tag{7}$$

where $Z_L \equiv X$, $z_L \equiv x$, and $z_{\ell-1} := G_\ell(z_\ell; u_\ell)$ for $\ell \leq L$. It is convenient to introduce the index notation $U_{>\ell} \equiv U_{\ell+1:L}$. First note that we can always write this posterior autoregressively as

$$p_{U_{1:L}|X}(u_{1:L}|x) = \prod_{\ell=1}^{L} p_{U_\ell|U_{>\ell},X}(u_\ell|u_{>\ell}, x). \tag{8}$$

Now, consider the graph of the full generative model shown in Figure 3a. It is clear that all paths from $U_\ell$ to any node in the set $\mathcal{H}_\ell := \{U_{>\ell}, Z_{>\ell}\}$ are *blocked* by $Z_\ell$, and hence that $Z_\ell$ d-separates $\mathcal{H}_\ell$ from $U_\ell$ (Bishop, 2006). Consequently,

$$p_{U_\ell|Z_\ell, U_{>\ell}, X}(u_\ell|z_\ell, u_{>\ell}, x) \equiv p_{U_\ell|Z_\ell}(u_\ell|z_\ell).$$

Informally,[6] we then have the following for any $\ell \in \{1, \ldots, L-1\}$:

$$
\begin{aligned}
p_{U_\ell|U_{>\ell}, X}(u_\ell|u_{>\ell}, x) &= \int p_{U_\ell, Z_\ell|U_{>\ell}, X}(u_\ell, z_\ell|u_{>\ell}, x) dz_\ell \\
&= \int p_{U_\ell|Z_\ell, U_{>\ell}, X}(u_\ell|z_\ell, u_{>\ell}, x) p_{Z_\ell|U_{>\ell}, X}(z_\ell|u_{>\ell}, x) dz_\ell \\
&= \int p_{U_\ell|Z_\ell}(u_\ell|z_\ell)\delta\left(z_\ell - \left(G_{\ell+1, u_{\ell+1}} \circ \cdots \circ G_{L, u_L}\right)(x)\right) dz_\ell \\
&= p_{U_\ell|Z_\ell}\left(u_\ell|\left(G_{\ell+1, u_{\ell+1}} \circ \cdots \circ G_{L, u_L}\right)(x)\right) \\
&= p_{U_\ell|Z_\ell}(u_\ell|z_\ell)
\end{aligned}
$$

by the definition of $z_\ell$, where we write $G_\ell(\cdot; u) \equiv G_{\ell, u}(\cdot)$ to remove any ambiguities when composing functions. We can substitute this result into equation 8 to obtain equation 7.

## B    DISCRETE $u$

While a discrete $u$ is appealing for its ability to produce exact likelihoods, it also suffers several disadvantages that we describe now. First, observe that the choice of the number of discrete values taken by $u$ has immediate implications for the number of disconnected components of $\operatorname{supp} p_X^\star$ that $G$ can separate, which therefore seems to require making fairly concrete assumptions (perhaps implicitly) about the topology of the target of interest. To mitigate this, we might try taking the number of $u$ values to be very large, but then in turn the time required to evaluate equation 4 (now a sum rather than an integral) necessarily increases. This is particularly true when using a stacked architecture, since to evaluate $p_X(x)$ with $L$ layers each having $K$ possible $u$-values takes $\Theta(K^L)$ complexity. Dinh et al. (2019) propose a model that partitions $\mathcal{X}$ so that only one component in each summation is nonzero for any given $x$, which reduces this cost to $\Theta(L)$. However, this partitioning means that their $p_X$ is not continuous as a function of $x$, which is reported to make the optimisation problem in equation 2 difficult.

Unlike for continuous $u$, the ELBO objective is also of limited effectiveness in the discrete case. Since $p_{U_\ell|Z_{\ell-1}}$ defines a discrete distribution at each layer, we would need a discrete variational approximation $q_{U_\ell|Z_\ell}$ to ensure a well-defined ELBO. However, the parameters of a discrete distribution are not ameanble to the reparametrisation trick, and hence we would expect our gradient estimates of the ELBO to have high variance. As mentioned above, a compromise here might be to use the CONCRETE distribution (Maddison et al., 2016) to approximate a discrete $u$ while still apply variational methods. We leave exploring this for future work.

## C    OTHER CHOICES OF INDEXED BIJECTION FAMILY

Other choices of $G$ than our suggestion in Subsection 3.4 are certainly possible. For instance, we have had preliminary experimental success on small problems by simply taking $g$ to be the identity, in which case the model is greatly simplified by not requiring any Jacobians at all. Alternatively, it is also frequently possible to modify the architecture of standard choices of $g$ to obtain an appropriate $G$. For instance, affine coupling layers, a key component of models such as RealNVP (Dinh et al., 2016), make use of neural networks that take as input a subset of the dimensions of $x$. By concatenating $u$ to this input, we straightforwardly obtain a family of bijections $G(\cdot; u)$ for each value of $u$. This requires more work to implement than our suggested method, but has the advantage of no longer requiring a choice of $s$ and $t$. We have again had preliminary empirical success with this approach. We leave a more thorough exploration of these alternative possibilites for future work.

---

[6]As with our arguments in Subsection 3.1, this could again be made precise using disintegrations (Chang & Pollard, 1997).

## D   FURTHER EXPERIMENTAL DETAILS

### D.1   LGF ARCHITECTURE

In addition to the bijections from the baseline flow, an LGF model of the form we consider requires specifying $p_{U_\ell|Z_{\ell-1}}, q_{U_\ell|Z_\ell}$, $s_\ell$ and $t_\ell$. In all our experiments

- $p_{U_\ell|Z_{\ell-1}}(\cdot|z_{\ell-1}) = \text{Normal}\left(\mu_p(z_{\ell-1}), \sigma_p(z_{\ell-1})^2\right)$, where $\mu_p$ and $\sigma_p$ were two separate outputs of the same neural network

- $q_{U_\ell|Z_\ell}(\cdot|z_\ell) = \text{Normal}\left(\mu_q(z_\ell), \sigma_q(z_\ell)^2\right)$, where $\mu_q$ and $\sigma_q$ were two separate outputs of the same neural network

- $s_\ell$ and $t_\ell$ were two separate outputs of the same neural network.

### D.2   2-D EXPERIMENTS

To gain intuition about our model, we ran several experiments on the simple 2-D datasets shown in Figure 1 and Figure 4. Specifically, we compared the performance of a baseline MAF against an LGF-MAF. We used MLPs for all networks involved in our model.

For the dataset shown in Figure 1, the baseline MAF had 10 autoregressive layers consisting of 4 hidden layers with 50 hidden units. The LGF-MAF had 5 autoregressive layers consisting of 2 hidden layers with 10 hidden units. Each $u_\ell$ was a single scalar. The $(s_\ell, t_\ell)$ network consisted of 2 hidden layers of 10 hidden units, and both the $(\mu_p, \sigma_p)$ and $(\mu_q, \sigma_q)$ networks consisted of 4 hidden layers of 50 hidden units. In total the baseline MAF had 80080 parameters, while LGF-MAF had 80810 parameters. We trained both models for 300 epochs.

We used more parameters for the datasets shown in Figure 4, since these targets have more complicated topologies. In particular, the baseline MAF had 20 autoregressive layers, each with the same structure as before. The LGF-MAF had 5 autoregressive layers, now with 4 hidden layers of 50 hidden units. The other networks were the same as before, and each $u_\ell$ was still a scalar. In total the baseline MAF had 160160 parameters, while our model had 119910 parameters. We trained all models now for 500 epochs.

The results of these experiments are shown in Figure 1 and Figure 4. Observe that LGF-MAF consistently produces a more faithful representation of the target distribution than the baseline. A failure mode of our approach is exhibited in the spiral dataset, where our model still lacks the power to fully capture the topology of the target. However, we did not find it difficult to improve on this: by increasing the size of the mean/standard deviation networks to 8 hidden layers of 50 hidden units (and keeping all other parameters fixed), we were able to obtain the result shown in Figure 5. This model had a total of 221910 parameters. For the sake of a fair comparison, we also tried increasing the complexity of the MAF model, by the size of its autoregressive networks to 8 hidden layers of 50 hidden units (obtaining 364160 parameters total). This model diverged after approximately 160 epochs. The result after 150 epochs is shown in Figure 5.

### D.3   UCI EXPERIMENTS

In Tables 3, 4, and 5, we list the choices of parameters for MAF and LGF-MAF. In all cases, we allowed the base MAF to have more layers and deeper coupler networks to compensate for the additional parameters added by the additional components of our model. Note that neural networks are listed as size $K_1 \times K_2$, where $K_1$ denotes the number of hidden layers and $K_2$ denotes the size of the hidden layers. All combinations of parameters were considered; in each case, there were 9 configurations for MAF and 8 configurations for LGF-MAF.

Table 3: Parameter configurations for POWER and GAS.

| | **Layers** | **Coupler size** | $u$ **dim** | $p, q$ **size** | $s, t$ **size** |
|---|---|---|---|---|---|
| MAF | 5, 10, 20 | $2 \times 100, 2 \times 200, 2 \times 400$ | - | - | - |
| LGF-MAF | 5, 10 | $2 \times 128$ | 2 | $2 \times 100, 2 \times 200$ | $2 \times 128$ |

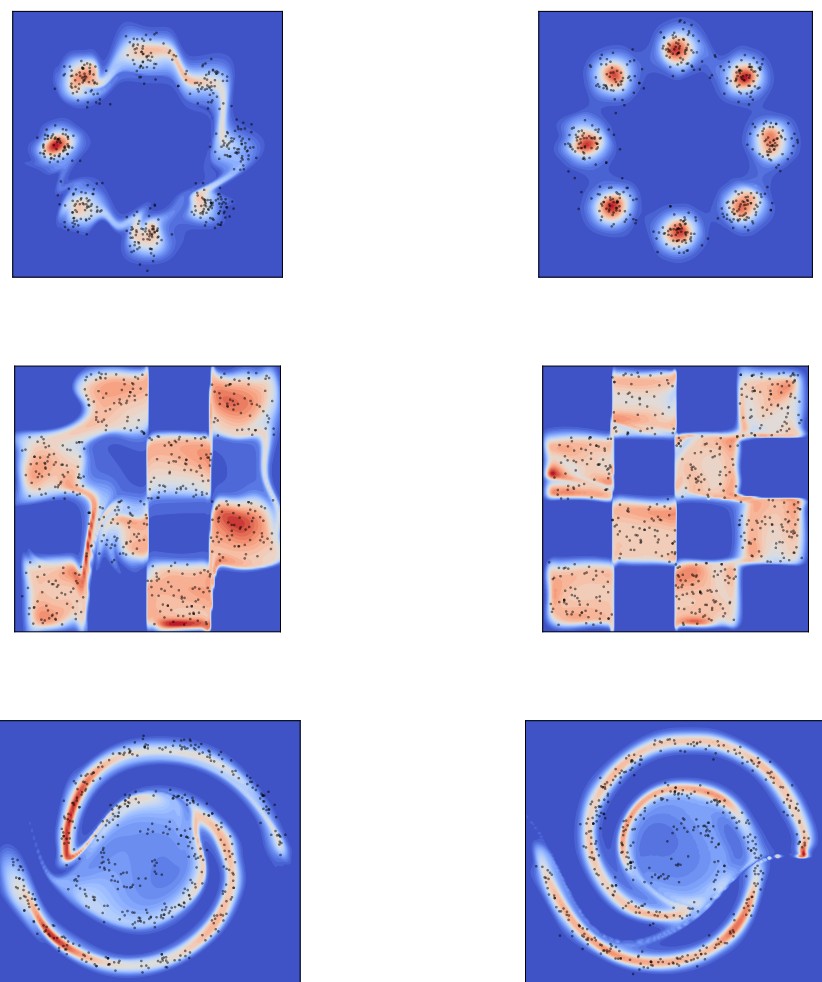

Figure 4: Density models learned by a standard 20 layer MAF (left) and by a 5 layer LGF-MAF (right) for a variety of 2-D target distributions. Samples from the target are shown in black.

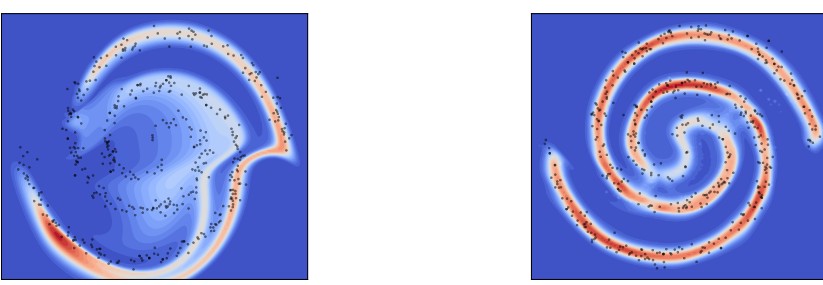

Figure 5: Density models learned by a larger 20 layer MAF (left) and a larger 5 layer LGF-MAF (right) for the spirals dataset.

Table 4: Parameter configurations for HEPMASS

|  | **Layers** | **Coupler size** | $u$ **dim** | $p, q$ **size** | $s, t$ **size** |
|---|---|---|---|---|---|
| MAF | 5, 10, 20 | $2 \times 128, 2 \times 512, 2 \times 1024$ | - | - | - |
| LGF-MAF | 5, 10 | $2 \times 128$ | 5 | $2 \times 128, 2 \times 512$ | $2 \times 128$ |

Table 5: Parameter configurations for MINIBOONE

|  | **Layers** | **Coupler size** | $u$ **dim** | $p, q$ **size** | $s, t$ **size** |
|---|---|---|---|---|---|
| MAF | 5, 10, 20 | $2 \times 128, 2 \times 512, 2 \times 1024$ | - | - | - |
| LGF-MAF | 5, 10 | $2 \times 128$ | 10 | $2 \times 128, 2 \times 512$ | $2 \times 128$ |

## D.4 IMAGE EXPERIMENTS

In figures 6 to 11, we present some samples synthesised from the density models trained on Fashion-MNIST and CIFAR-10.

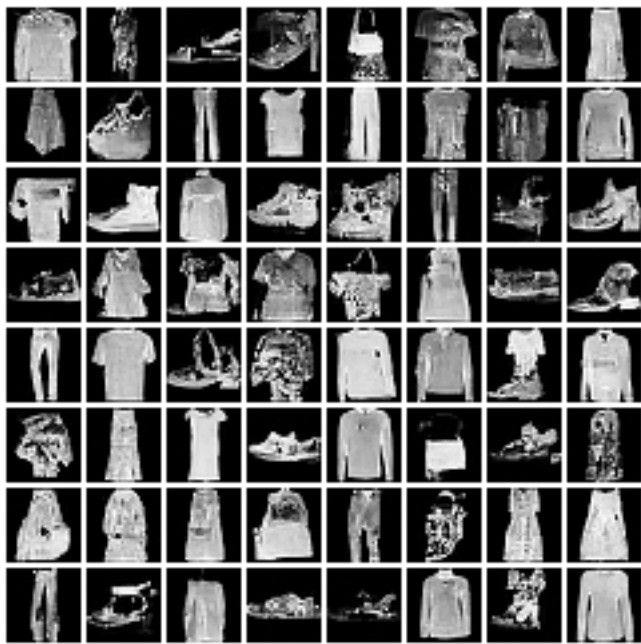

Figure 6: Synthetic samples from Fashion-MNIST generated by RealNVP (4)

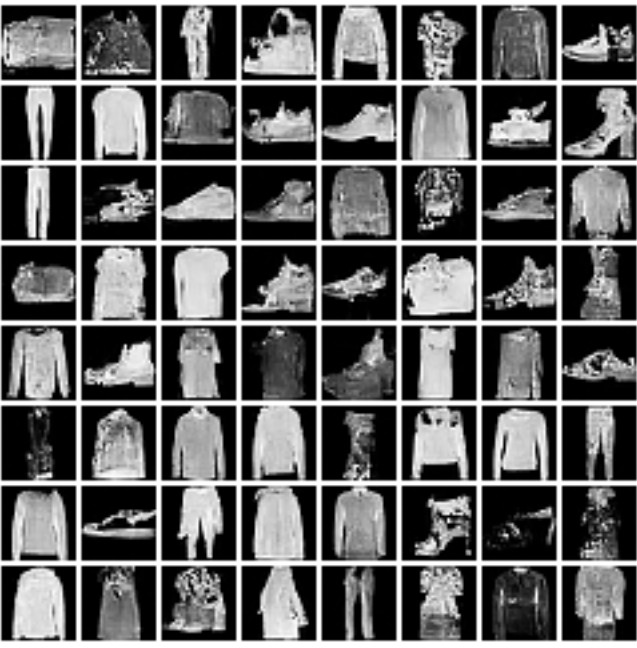

Figure 7: Synthetic samples from Fashion-MNIST generated by RealNVP (8)

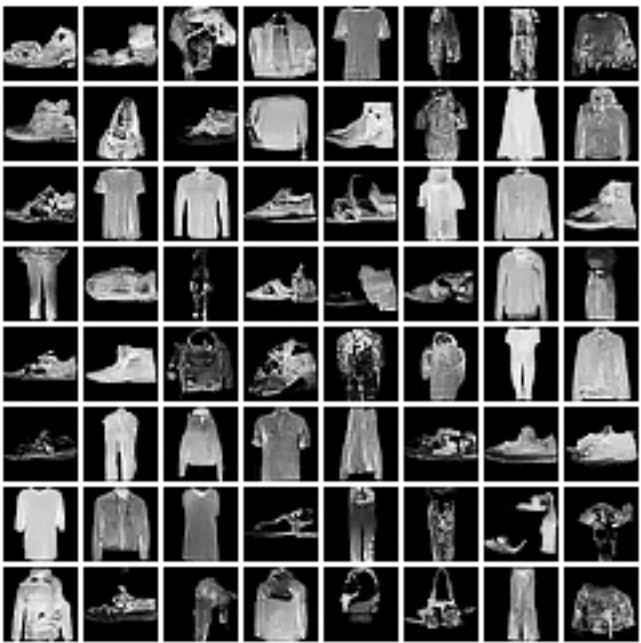

Figure 8: Synthetic samples from Fashion-MNIST generated by LGF-RealNVP (4)

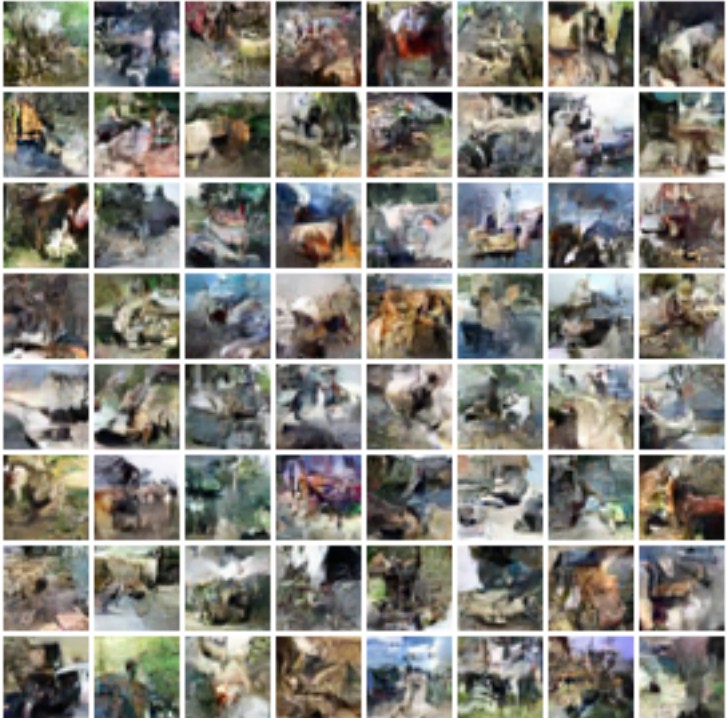

Figure 9: Synthetic samples from CIFAR-10 generated by RealNVP (4)

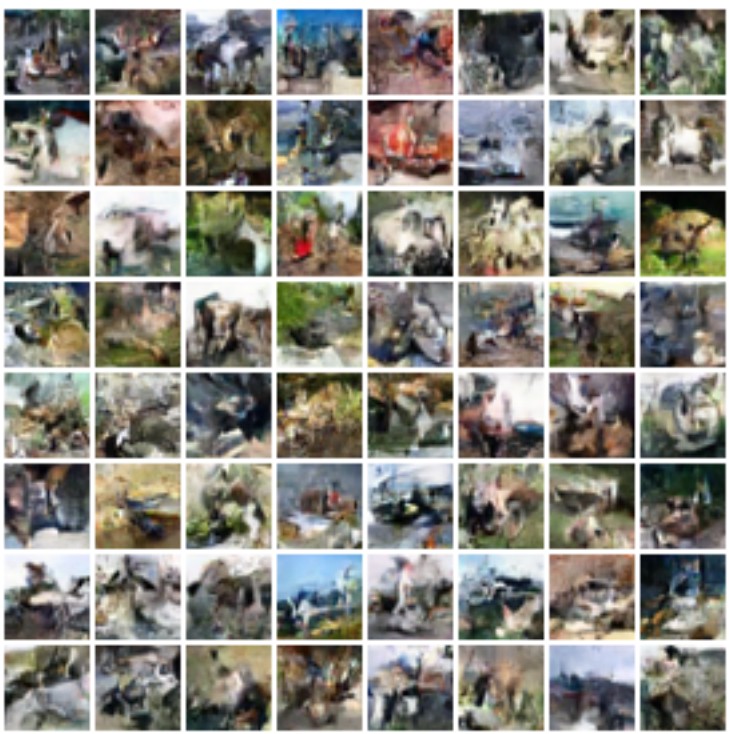

Figure 10: Synthetic samples from CIFAR-10 generated by RealNVP (8)

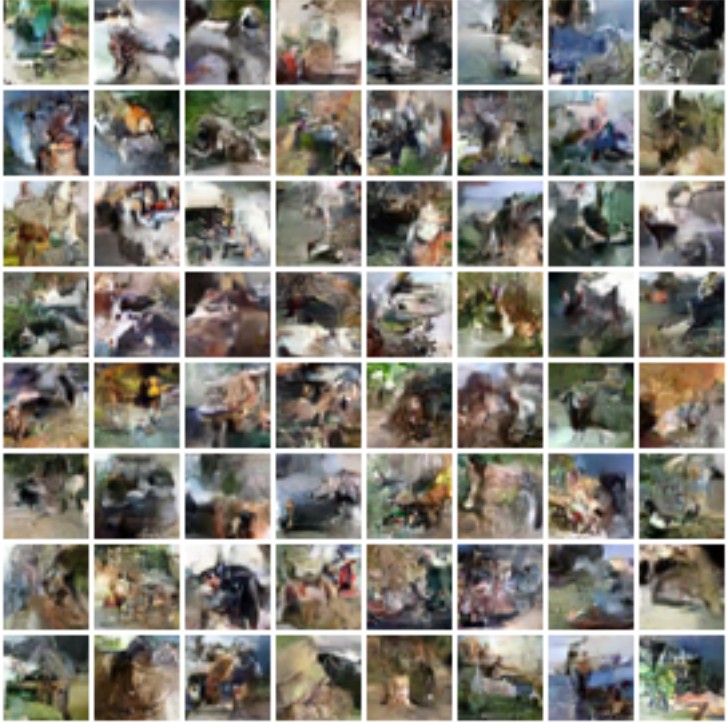

Figure 11: Synthetic samples from CIFAR-10 generated by LGF-RealNVP (4)

