# OpenReview forum: "Localised Generative Flows"
_ICLR.cc/2020/Conference — Reject_

### Official Review · AnonReviewer2 · 2019-10-17
**Official Blind Review #2**

**Rating:** 3

**Review:**

The paper introduces a straight-forward way to expand the flow models by considering mixture of flow distributions. The idea is not very novel since several previous work have tried the mixture of flow such as the mentioned RAD and Deep Mixture. The paper studies some further improvements such as using the continuous auxiliary variable and stacking multiple mixture layers.

The major concerns are the following:

1) The paper tries to solve a “problem” build upon intuition. The paper explains as “the normalizing flow places global constraint on the bijection”, “it need to match the topology of  X to the topology of Z ”, “continuous functions necessarily preserve topology”. What kind of topological properties are referred to here? Are all topological properties preserved under continuous function? It needs to be more accurate when using such terminologies. The intuition of the paper is weak and heuristic. The example in Figure 1 can potentially be easily solved with a two component Gaussian mixture of input Z to a vanilla flow model.

2) For the mixture p(X), will the proposed method generate samples concentrated on one or some of the components? Why or why not?

3) Considering there are a plenty of improvements of flow models, it is neccesary for the proposed method to compare with, at least for some methods explained in the Related Work section.

4) Since the proposed methods inevitably lose the advantage of analytic density property of flow methods, it is better to show some advantage over implicit or semi-implicit methods. For example, (https://arxiv.org/abs/1805.11183) also uses a hierarchical model with continuous auxiliary variables and a marginalization similar to Eq.(4) in this paper. How does the proposed method related to or compared to these methods?

5) In experiment section, in Table 1, the proposed method is worse than MAF for 2 datasets out of 4? In Table 2, what are the numbers refer to?

6) On page 4, it should be p(U|Z) not p(U|X)?

In sum, I think though the paper makes contribution on exploring better flow models but the novelty is relatively weak, the discussion and comparison of related work is insufficient and the experiments are not convincing or have mistakes. I think a modification is necessary before publishing.

################

I have read the author's feedback.



**Experience Assessment:**

I have published in this field for several years.

**Review Assessment: Checking Correctness Of Derivations And Theory:**

I assessed the sensibility of the derivations and theory.

**Review Assessment: Checking Correctness Of Experiments:**

I carefully checked the experiments.

**Review Assessment: Thoroughness In Paper Reading:**

I read the paper thoroughly.

---

> ### Author Response · Authors · 2019-11-15
> **Response to Official Blind Reviewer #2**
>
> Thank you for your review.
>
> We first address the claim that LGFs lack novelty. Please see above our discussion about RAD. In particular, our use of continuous mixing variables is not simply an incremental design choice, but provides a method for circumventing the computational cost which is exponentially increasing with the depth of a standard deep mixture model (as mentioned in the Appendix and related work section) _without_ introducing discontinuities into the loss function or explicit partitioning schemes. Moreover, continuous mixing variables require a significantly different approach to training (namely a variational scheme) which further distinguishes LGFs from these other methods.
>
>
> Please see below our responses to your other points:
>
> 1) Regarding the claim that our paper is pure intuition, please see the discussion above. As we discuss there, although we did not previously formulate our discussion in terms of concrete mathematical statements, we did take care to be precise where this was possible. In particular, note that when we talk about "preserving topology", we mean so in the well-defined mathematical sense that the topology (i.e. the open sets) of a space is preserved under continuous mappings that have continuous inverses. Such functions indeed preserve _every_ topological characteristic, including properties such as connectedness, compactness, genus (i.e. the number of "holes" in the space), etc. These are standard mathematical concepts, and we refer you to https://en.wikipedia.org/wiki/Homeomorphism for more information and references.
>
> Regarding the comment that the example in Figure 1 would be fixed by a simple mixture model - we acknowledge this directly in the 4th paragraph of section 2. However, as we argue there, this sort of approach does not scale to complicated datasets (like CIFAR10) where the topology of the target is completely opaque. In these instances we would like a method that can somehow learn the topology of the target on its own.
>
> 2) As we discuss in section 2, the maximum likelihood objective corresponds to a mode-covering KL objective. In other words, the loss function is encouraged to ensure the support of our model covers all the modes of the target. This is a standard feature of likelihood-based training and is not specific to our model.
>
> 3) Refer to the general comment above for a discussion on testing the improvement LGFs make on other flow models. As for other variational improvements on flow models, those are discussed in the related work section. These methods are either outside of the scope of LGFs (e.g. Das et al., (2019), Gritsenko et al. (2019)), orthogonal improvements that could be combined with LGFs (e.g. Ho et al. (2019)), or RAD (Dinh at al., 2019) which is discussed above.
>
> 4) The linked work (Semi-Implicit Variational Inference (SIVI)) may indeed seem superficially similar, but it is notably different in both its motivation and application. In particular, SIVI is a method which looks to improve the approximate variational distribution in variational inference; LGF is a method which looks to improve the bijection approach in density estimation using normalizing flows. LGFs require a variational scheme to optimize their parameters (for which methods like SIVI could potentially be useful), but this is different from considering LGFs to be a variational method on their own. Plus SIVI has no notion of stacking models to obtain even greater expressiveness - there is only one hierarchical layer.
> GANs are another instance of an implicit model, but they provide no reliable method to approximate log-likelihoods.
>
> 5) It is not clear what is referred to here. Note that we report the average test set log-likelihood, for which higher is better. In all cases in Table 1, the log-likelihood is higher for LGF-MAF than for MAF.
>
> 6) Thank you for pointing this out. We have corrected this typo.
>
> Finally, we wonder whether you can comment further on the mistakes you perceive in our experiments? Please note point 5) above.

---

### Official Review · AnonReviewer3 · 2019-10-18
**Official Blind Review #3**

**Rating:** 3

**Review:**

The authors propose to extend flow-based density models by replacing a single bijection with a hierarchical mixture of bijections. Each component in the mixture is then required to only push the prior onto a local region; this helps improve the coverage of $\mathcal{X}$. This is motivated by the conjecture that in many cases the topology of $\mathcal{X}$ might be overly complicated to be effectively captured by a single bijection. Formally, this is achieved by introducing a conditional random variable $U|Z$. In doing however, the log-likelihood is rendered intractable and a variational approximation must instead be resorted to. A recursive formula for computing the ELBO is introduced in this vain.

Overall, I think this is a generally interesting contribution to the normalizing-flow literature that I expect to spark further research. However, there are some rough edges to this paper. The initial motivation is well-presented and relatively easy to follow, though a diagram would serve to cement the intuition regarding the support mismatch. The issue mentioned in footnote 2 deserves further discussion. At the same time, while well-reasoned, their justifications are nonetheless largely conjectural and further theoretical or empirical evidence would be welcome, both for characterising the pathologies they aim to redress and their proposed solution.

For the experiment section, I would have liked to have seen comparisons not only to the simplest baseline but also to some of the other methods mentioned in related works. In general, the experiment section is quite short and I didn't get a very good sense of how well this method performs.

The following should be addressed:

- provide more evidence for the conjectures surrounding the motivation and derivation
- supply more varied baselines (e.g. RAD model)

Minor comments:

- at the top of page 4, you refer to $p_{U|X}$ several times, but I think you mean $p_{U|Z}$
- in the paragraph after equation (2), the $\theta$ superscript seems to be missing from the $p_X^\theta$
- in the first sentence of section 3.1, you refer to "the single $g$ used in equation 2", but equation 2 mentions no $g$
- in the first line on page 3, you talk about some region of supp $p_X^\theta$ being pushed out of supp $p_X^*$, shouldn't this be the other way round since the KL is infinite only if the the support of $p_x^*$ is not contained within the support of $p_X^\theta$?

**Experience Assessment:**

I have read many papers in this area.

**Review Assessment: Checking Correctness Of Derivations And Theory:**

I assessed the sensibility of the derivations and theory.

**Review Assessment: Checking Correctness Of Experiments:**

I carefully checked the experiments.

**Review Assessment: Thoroughness In Paper Reading:**

I read the paper at least twice and used my best judgement in assessing the paper.

---

> ### Author Response · Authors · 2019-11-15
> **Response to Official Blind Reviewer #3**
>
> Thank you for your review.
>
> Regarding your points about the theoretical underpinnings and empirical evaluation of our method, please see our general replies in the Common thread above.
>
> Thank you very much for your other feedback also. We agree a diagram could help to convey intuition. We have also fixed the typos and clarity issues that you have pointed out in the uploaded version of the paper.

---

### Official Review · AnonReviewer1 · 2019-10-24
**Official Blind Review #1**

**Rating:** 1

**Review:**

Summary:
This paper conjectures that normalizing flows are fundamentally limited due to the architecture assumption that the generative function g is continuous in x. It is argued that this constraint makes maximum likelihood estimation difficult in general. Localised generative flows are proposed as a solution and consist in modeling the generative model as a continuous mixture of bijections. Experiments suggest an improvement over MAF.

Decision:
The observation that continuity imposes a hard constraint on the network is sound, and the proposed solution appears to show some improvement. However, in its current state, this work appears to be quite fragile both from a theoretical and experimental point of view. First, it is only conjectured that this constraint poses actual problems. Second, the experimental evaluation is weak and insufficient. It omits comparisons with more recent generative flows that have shown to be able to model discontinuous densities. For this reason, I do not recommend the paper for acceptance.

Further arguments:
- The whole paper rests on intuition without strong theoretical backup.
- The experiments are quite poor and results frankly oversold. It is said the method "improves performance across a variety of common density benchmarks". While we see improvements in Table 1 over MAF, the comparison omits all recent architectures based on Normalizing Flows, such as TAN (Olivia et al, 2018), NAF (Huang et al, 2018), B-NAF (De Cao et al, 2019) or SOS (Jaini et al, 2019). All of those methods have reported better results than those provided in Table 1. They have also been shown empirically to work for discontinuous densities. While I understand that LGF can be combined with any flow architecture, the question remains whether using a continuous mixture translates into significant improvements for those baselines as well. The experimental benchmarks also omit datasets such as BSDS300, for which the higher dimensionality is usually challenging. The same goes for Table 2 which omits recent and better results, such as Glow or FFJORD.
- Closer to LGF, a proper experimental comparison to RAD (Dinh et al, 2019) would be appreciated.
- The proposed architecture supposedly enables better generative models. However, this comes at the price that the density can no longer be evaluated exactly and analytically. Since normalizing flows are also typically slow for sampling, this makes the benefits of the proposed architecture quite limited. In particular, it is not clear why generative models that are good at sampling only (e.g., GANs) should not then be preferred?
- As a result of the point above, the experimental results are reported only in terms of approximated negative log-likelihood. I do not think this is fair, since models like MAF do provide exact values. It also makes the comparison with previous methods more difficult.

Further feedback:
- As per ICLR policy, higher standards should be applied to papers with 9 or more pages. I am confident the paper could be written within 8 pages only.


**Experience Assessment:**

I have published one or two papers in this area.

**Review Assessment: Checking Correctness Of Derivations And Theory:**

I assessed the sensibility of the derivations and theory.

**Review Assessment: Checking Correctness Of Experiments:**

I carefully checked the experiments.

**Review Assessment: Thoroughness In Paper Reading:**

I read the paper at least twice and used my best judgement in assessing the paper.

---

> ### Author Response · Authors · 2019-11-15
> **Response to Official Blind Reviewer #1**
>
> Thank you for your review.
>
> First and foremost, we wish to state our objection to certain uncharitable comments made within this review. Regarding your claim that our experiments are oversold -- we respectfully disagree with this. The statement of ours that you quote describes improved performance on a variety of _tasks_. We stand by this claim: compared with the standard flow-based baselines that we consider, we found that LGFs do improve density estimation for 2D densities, real-world tabular data, and high-dimensional image data, all of which have very distinct structures and dimensionalities. Please see our general reply above for further discussion of this point, as well as further empirical results.
>
> In a similar vein, when describing our contribution, you state that our "(e)xperiments suggest an improvement over MAF". We again refer you to our discussion above, and emphasise that, for the models considered, our results demonstrate that our method yields significant and unambiguous benefit. Moreover, in addition to MAF, which we apply to tabular data, we also consider a large-scale RealNVP model that makes use of fully convolutional networks and a multi-scale architecture. We believe these changes yield a density model with a significantly different structure and characteristics to MAF, and it bears emphasising that our method provides benefit within this quite distinct context also.
>
> In response to your claim that our paper is purely based on intuition, we refer you to our general reply above.
>
> Regarding BSDS300 - although we omitted this benchmark, we did consider Fashion-MNIST and CIFAR10, both of which have far higher dimensionality than this dataset (784 and 3072 as opposed to 63 dimensions). Even with this increase in dimension, LGFs still yielded a performance benefit over the baselines we considered. Please also see above for our results using Glow that we have obtained subsequently.
>
> Regarding RAD - please see above.
>
> Finally, regarding your points about the inexact log-likelihoods: as we argue in section 3.3.1 of the paper, this does not pose a major limitation for our method. When log-likelihoods are required at evaluation time, it is straightforward to obtain an estimate using importance sampling as described. This estimate is consistent in the sense that it is possible to achieve as much accuracy as desired simply by taking more importance samples. This approach is standard, for example, within the VAE literature -- and indeed your comment would seem to apply equally to all of these models as well as ours.
>
> For implicit models like GANs, no straightforward estimate of the likelihood is available at all. In this setting it is also typically impractical to estimate the latent distribution $p(z|x)$ for a given $x$, which can be useful for downstream tasks. We also mention that various normalising flow models exist that do provide fast sampling as well as density estimation -- for instance, RealNVP, which we also consider in this paper.
>
> Finally, we have uploaded a revision of the paper that is not longer than 8 pages.

---

### Public Comment · ~Kevin_Zhang2 · 2019-09-27
**No code in provided github link even after 56 hours of submission deadline**

Hi,
As of close to 56 hours after submission deadline , no code is present in the provided github link. It is not fair to provide a placeholder link for code submissions (which impact the review process) and submit code taking considerable buffer time after submission deadline.

---

> ### Author Response · Authors · 2019-10-01
> **Response**
>
> Hi Kevin,
>
> We are working hard to release our code as quickly as possible, and at this stage plan to do so by October 4th, when reviewers will be assigned to papers.
>
> We respectfully disagree with your assessment that this practice is unfair, since the facility to upload code past the submission deadline is available to all authors, either via the method we have chosen, or by posting an updated link as a comment to their work.
>
> We also note that there is nothing in the ICLR call for papers that forbids or even discourages this approach.
>
> We will be fully transparent about the exact times that we upload our work, which will be timestamped in our git commits. We leave any decisions about how to make use of this information to the discretion of the reviewers.

---

### Author Response · Authors · 2019-11-15
**Common Response to All Reviewers - Part I**

We thank the reviewers for their feedback and comments. In this thread, we respond to several points that were common across all reviews. More specific replies are also given in the individual reviewer threads below.

1. Is our paper purely conjectural?

All reviewers were critical of what they perceive to be a lack of solid theoretical justification for our approach. It is indeed true that our discussion involves some conjecture, which we are careful to make explicit whenever it occurs. However, many of our statements are mathematically precise. For instance, the discussion in the second paragraph of section 2 constitutes a proof of the following proposition:

* Proposition: If the supports of $p^*_X$ and $p_Z$ are not homeomorphic, then no normalising flow model with $p_Z$ as the prior will yield $p_X^*$ exactly.

This proposition pinpoints a concrete, well-defined problem with normalising flows that, to our knowledge, has not been addressed in the literature to date.

Similarly, although presented informally, our discussion in section 3.2 of the paper also provides a clear theoretical account of the potential benefits of LGF models compared with standard normalising flows. This discussion can be summarised by the following proposition:

* Proposition: Suppose that $\text{supp} p_X^*$ is open and $G(\cdot; u)$ is continuous for each $u$. Suppose further that, for each $x \in \text{supp} p_X^*$, the set
\[
    B_x := \{u : p_Z(G(x; u)) |\det DG(x; u)| > 0 \}
\]
has positive Lebesgue measure. Then there exists $p_{U|Z}$ such that $\text{supp} p_X = \text{supp} p_X^*$.

In other words, a sufficiently expressive $p_{U|Z}$ can correct the problem identified in the earlier proposition, which stems from the fact that homeomorphisms do not exist between sets with different topologies. The condition on $B_x$ is trivially satisfied in the standard case that $p_Z$ has full support and $|\det DG(x;u)| > 0$ for all $x$ and $u$. The proof of this result is straightforward and proceeds exactly as described in section 3.2 of our paper, with $p_{U|Z}$ constructed so as to downweight regions of mass that fall outside the support of the target.

To better clarify our discussion in the paper, we have stated both results more formally in an updated version resubmitted above.

[Continued below....]

---

> ### Author Response · Authors · 2019-11-15
> **Common Response to All Reviewers - Part II**
>
> 2. Were our experiments comprehensive enough?
>
> All reviewers argued that our experiments were not comprehensive enough to serve as a justification for our method, suggesting that, while we demonstrate improvement on the baseline models we consider, we may encounter diminishing returns when using LGFs in conjunction with a stronger underlying model.
>
> To this, we first emphasise that the experiments we report do indeed demonstrate a conclusive and comprehensive benefit of LGFs within the scope considered. For the MAF and RealNVP models that we use, LGFs _uniformly_ improve performance across a variety of interesting density tasks. These tasks vary significantly in size, with the dimensionality of the data ranging over 3 orders of magnitude from fewer than 10 dimensions (for the 2D datasets, GAS, and POWER) to 3072 dimensions (for CIFAR-10). These tasks also vary significantly in terms of structure; in particular, we show that LGFs scale to handle large-scale, fully convolutional, multi-scale density models (i.e. the RealNVP model considered) that exploit the pixel structure of MNIST and CIFAR-10. (Indeed, this has not been demonstrated for several of the benchmarks suggested in the reviews.) While we do not include results for current state-of-the-art density models, we believe the consistent improvement over these baselines alone is compelling.
>
> However, we do also agree that more baselines are always better. To this end, we report additional experimental results here. First, we considered the Rational Quadratic Spline from the Neural Spline Flows (NSF) paper (Durkan et al., 2019). Here, LGF yielded improved stability and test scores for a variety of 2D experiments, which can be reproduced using an update that we have pushed to our repository above. (See for example the "rings" dataset, which is quite topologically distinct from the standard Gaussian prior and therefore difficult to train on for the baseline.) We also compared NSF with LGF-NSF on the POWER, GAS, and MINIBOONE UCI datasets (the limited time for rebuttal precluded the inclusion of the HEPMASS dataset). For each dataset and model, we report the test-set log likelihood averaged over three random seeds (higher is better). We use the hyperparameter settings provided in the Appendix of the NSF paper for the baseline. On POWER: NSF achieves 0.65, with 3004722 parameters; LGF-NSF achieves 0.66, with 3064418 parameters. On GAS: NSF achieves 12.50, with 3128392 parameters; LGF-NSF achieves 12.29, with 3161104 parameters. On MINIBOONE: NSF achieves -10.80, with 212102 parameters; LGF-NSF achieves -9.51, with 169168 parameters. So, when both models have a number of parameters that is approximately equal, we obtained the same behaviour on POWER, slightly lower likelihoods on GAS, but a significant improvement on MINIBOONE. We also note that we achieved these results with the first hyperparameter settings (namely, the choice of $p$, $q$, and $s$ and $t$ networks) for our model that we attempted, and therefore expect further improvements to be possible here.
>
> We additionally experimented with a large-scale Glow model (Kingma & Dhariwal, 2018). As a baseline, we used exactly the model used in (Kingma & Dhariwal, 2018), but replaced their act norm layers with batch norm. We made our LGF model shallower, using only 2 multi-scale steps as opposed to the default of 3. We also reduced the number of hidden channels in the coupling networks from 512 (the default) to 256. Both our baseline and LGF here achieve the same 3.40 BPD on CIFAR10 after 1000 epochs. However, our model uses only 15M parameters, as opposed to the 44M required by Glow. Again, we achieved these results with minimal tuning of our method, and we believe it possible to improve performance here even further.
>
> We also tried SoS (Jaini et al, 2019), but found we could not reproduce the results described in the paper. In particular, we found the method to be unstable -- to the extent that, for the UCI datasets, we did not achieve a non-NaN value for the average test-set log likelihood on any run (i.e. at least one test value was NaN). We believe this is due to the fact that the suggested configuration in Jaini et al. (2019) means the bijection considered involves 9th order polynomials, which can produce extremely large values when given inputs with values outside the range $[-1, 1]$. It was therefore unclear how to incorporate LGFs into this context.
>
> [Continued below....]

---

> > ### Author Response · Authors · 2019-11-15
> > **Common Response to All Reviewers - Part III**
> >
> > We likewise considered B-NAF (De Cao et al, 2019) model, but also encountered problems. In particular, we note that the default choice of $\tanh$ nonlinearity suggested in the paper (and which we believe their results are based on) means that their model is not a bijection, since $\tanh$ is not surjective. The suggested alternative choice of LeakyReLU does fix this, but then introduces a second problem of removing the gradient signal on the Jacobian terms in the loss, which become constant almost everywhere. We sought to address this by considering a soft version of the LeakyReLU defined by $x \mapsto \epsilon x + (1 - \epsilon) \log(1 + e^x)$, where $\epsilon \in (0, 1)$  corresponds to the slope on the negative part of the real line. However, while improving over the "hard" LeakyReLU, this did not train successfully even for simple 2D experiments. It was again unclear for us how best to consider an LGF version of B-NAF.
> >
> > We have updated our repository with code to reproduce all these results, including for SoS and B-NAF.
> >
> > 3. Why didn't we explicitly compare against RAD?
> >
> > All reviewers cited RAD (Dinh et al., 2019) as a particular benchmark that we ought to have compared against. We agree that RAD is a very interesting paper. However, we do not believe that the version of RAD that exists online at present is ready to be used as a benchmark yet.
> >
> > The choice of a discrete mixing variable brings significant difficulties that we discuss in section B of our appendix. In particular, naive stacking entails that the cost of evaluating likelihoods grows exponentially in the depth of the model, which quickly becomes intractable as larger models are used e.g. for image datasets.
> >
> > RAD avoids this by partitioning the input space in a way that achieves a linear cost in the depth of the model. However, this partitioning scheme means that the loss landscape becomes discontinuous. Some guidance is provided in their appendix on how to resolve this problem for the case of a one-dimensional density with three components, but it is not immediately clear how to extend this to higher dimensions or more complicated targets. It therefore fell outside the scope of this paper to establish RAD as a comparable benchmark for the problems we wished to consider. We also note that the RAD paper itself does not consider problems in higher than 2 dimensions.

---

### Decision · Program_Chairs · 2019-12-19

**Decision:**

Reject

**Comment:**

This paper proposes to overcome some fundamental limitations of normalizing flows by introducing auxiliary continuous latent variables. While the problem this paper is trying to address is mathematically legitimate, there is no strong evidence that this is a relevant problem in practice. Moreover, the proposed solution is not entirely novel, converting the flow in a latent-variable model. Overall, I believe this paper will be of minor relevance to the ICLR community.